# Structural and Functional Characterization of the Vacuolar-Type Na^+^, K^+^/H^+^ Antiporter NHX1 from Rice (*Oryza sativa* L.)

**DOI:** 10.3390/biom15111513

**Published:** 2025-10-27

**Authors:** Boning Cao, Zhiyong Xing, Jingxian Li, Ying Yuan, Xueru Lei, Hong Tang, Dan Wang, Jiali Ma, Shuangping Heng, Lin Cheng

**Affiliations:** 1Henan International Joint Laboratory of Tea-Oil Tree Biology and High Value Utilization, College of Life Sciences, Xinyang Normal University, Xinyang 464000, China; caobn@xynu.edu.cn (B.C.);; 2Dabie Mountain Laboratory, College of Tea and Food Sciences, Xinyang Normal University, Xinyang 464000, China

**Keywords:** Na^+^, K^+^/H^+^ antiporters, salt stress, homology modeling, AlphaFold, ion transport

## Abstract

Plant vacuolar-type Na^+^, K^+^/H^+^ antiporters (NHXs) play important roles in pH and K^+^ homeostasis and osmotic balance under normal physiological conditions. Under salt stress, vacuolar-type NHX enhances salt tolerance by compartmentalizing Na^+^ into vacuoles. However, the ion transport mechanism of vacuolar-type NHX remains poorly understood due to the absence of resolved protein crystal structures. To investigate the ion transport mechanism for vacuolar-type NHX, the three-dimensional structure of rice vacuolar-type NHX1 (OsNHX1) was established through homology modeling and AlphaFold3.0. The OsNHX1 model contains thirteen transmembrane segments according to hydrophobic characteristics and empirical and phylogenetic data. Furthermore, this study validated the OsNHX1 model via functional experiments, revealing a set of key charged amino acids essential for its activity. Mapping these amino acids onto the OsNHX1 model revealed that its pore domain exhibits a transmembrane charge-compensated pattern similar to that of NHE1 while also displaying a distinct charge distribution on either side of the pore domain. Comparative analysis of the key amino acid sites responsible for ion transport in the crystal structure of OsSOS1 and NHE1 revealed that OsNHX1 employs a unique ion transport mechanism. This study will enhance our understanding of the function and catalytic mechanism of OsNHX1 and other plant vacuolar-type NHXs.

## 1. Introduction

Soil salinization is widely acknowledged as a critical environmental issue with profound implications for the biosphere and ecological systems. Saline–alkali soils are globally prevalent, occupying an estimated 1 billion hectares of land [1]. Soils are classified as salt-affected when their salt concentrations exceed levels that are toxic to plant growth. When exposed to salt stress, plants experience excessive accumulation of Na^+^ ions within their cells, which disrupts cellular pH, osmotic pressure, and ion homeostasis. These disturbances ultimately result in metabolic dysfunction, hindering normal growth and development. To counteract the cytotoxic effects of elevated Na^+^ concentrations, plant cells employ adaptive mechanisms such as Na^+^ efflux and compartmentalization, which serve to lower cytoplasmic Na^+^ levels and restore intracellular stability. Studies have identified plant Na^+^, K^+^/H^+^ antiporters (NHXs) as key regulators of salt tolerance. These transporters belong to the NHE/NHX subfamily within the CPA (cation/proton antiporter) family, a group of eukaryotic ion transport systems [2]. The monovalent cation–proton antiporter (CPA) superfamily comprises a highly conserved group of transmembrane proteins that facilitate the exchange of cations with protons (H^+^) in opposite directions. This exchange plays a critical role in regulating the local pH, electrical gradients, and cation homeostasis, underscoring their fundamental importance in maintaining cellular and organismal homeostasis by controlling monovalent ions such as sodium (Na^+^), potassium (K^+^), and protons (H^+^). The CPA superfamily is further classified into two distinct families: CPA1 and CPA2 [3,4,5,6].

Plant Na^+^, K^+^/H^+^ antiporters belong to the NHE/NHX subfamily of the eukaryotic ion transport system CPA1 family [7,8]. Based on subcellular localization and phylogenetic analysis, they can be classified into three categories: vacuolar membrane-type NHX (Class I), endomembrane-type NHX (Class II), and plasma membrane-type NHX (Class III). Taking the Arabidopsis *NHX* gene family (AtNHX1–8) as an example, AtNHX1–4 (Class I) are localized to the vacuolar membrane. Under normal physiological conditions, they primarily regulate cellular pH, K^+^ homeostasis, and osmotic balance, playing crucial roles in growth and development. Under salt stress, they compartmentalize Na^+^ into vacuoles and modulate cytoplasmic K^+^ homeostasis to maintain ion balance, thereby enhancing plant salt tolerance [9,10,11,12]. AtNHX5–6 (Class II), which exhibit functional redundancy, co-localize to the Golgi apparatus, trans-Golgi network (TGN), and prevacuolar compartment (PVC). As endomembrane-type NHX proteins, they are essential for maintaining pH and K^+^ balance in the endomembrane system, playing critical roles in protein processing, sorting, and the regulation of fruit and root growth [13,14,15,16]. AtNHX7 (also known as SOS1, Salt Overly Sensitive 1) and AtNHX8 (Class III) are localized to the plasma membrane. AtNHX7 primarily mediates Na^+^ efflux and is indispensable for plant salt tolerance [17], while its homolog AtNHX8 mainly transports Li^+^ [18]. In rice, the *NHX* gene family has six members, including *OsNHX1–4* (vacuolar membrane-type NHX), *OsNHX5* (endomembrane-type NHX) and *OsSOS1* (plasma membrane-type NHX) [19].

The NHX/NHE protein consists of an N-terminal region containing 10–13 highly conserved, hydrophobic transmembrane domains that constitute the Na^+^/H^+^ exchanger module, which facilitates the transport of Na^+^ and K^+^. Conversely, the hydrophilic C-terminal region plays a regulatory role in modulating the transport activity of these ions [3,14,20,21,22]. To date, the crystal structures of NHX proteins from both bacterial and eukaryotic sources have been successfully resolved. These include EcNhaA from *Escherichia coli* [23,24], TtNapA from *Thermus thermophiles* [25], MjNhaP1 from *Methanococcus jannaschii* [26], PaNhaP from *Pyrococcus abyssi* [27], and mammalian NHA2 [28] and NHE1 [29] from humans, as well as plant plasma membrane-type NHX proteins such as AtSOS1 from *Arabidopsis thaliana* [30] and OsSOS1 from *Oryza sativa* [31]. Among them, MjNhaP1, PaNhaP, NHE1, AtSOS1, and OsSOS1 belong to the CPA1 family, while EcNhaA, TtNapA, and NHA2 belong to the CPA2 family. Topological analysis reveals that the CPA2 proteins EcNhaA and TtNapA each possess 12 transmembrane domains. In contrast, the NHA2 protein contains 14 transmembrane domains. Meanwhile, the CPA1 proteins MjNhaP1, PaNhaP, NHE1, AtSOS1, and OsSOS1 each exhibit 13 transmembrane domains. The structural analysis of these proteins indicates significant differences in ion transport mechanisms between different classes of NHX/NHE proteins.

Currently, the ion transport mechanism of plant vacuolar-type NHX remains poorly understood due to the absence of resolved protein crystal structures. Furthermore, there is still controversy surrounding the topological structure model of vacuolar NHX. The topological models of plant vacuolar-type NHX were described for the AtNHX1 [20,32]. Yamaguchi et al. proposed a topological model of AtNHX1 that contains 12 transmembrane segments, with N- and C-termini located at the cytoplasmic side and vacuolar lumen side, respectively, applying epitope tagging and protease protection assays in the yeast heterologous expression system. Sato and Sakaguchi proposed a topological model of AtNHX1 that contains 12 transmembrane segments, with N- and C-termini located at the cytoplasmic side. This model was developed based on in vitro analysis of various fragments of the AtNHX1 protein. The elucidation of crystal structures in various NHX/NHE proteins has significantly advanced our understanding of the mechanisms underlying conformational changes, ion binding and transport, and ion selectivity in these transport proteins. This progress has also facilitated the identification of several functional amino acid sites. Given the high conservation of the Na^+^/H^+^ antiporter family across microorganisms, plants, and animals, homology modeling can effectively identify candidate amino acid sites involved in ion binding and transport within plant vacuolar-type NHX. Additionally, the public availability of AlphaFold and its associated predicted protein structure database has further improved the accuracy of protein structure predictions [33,34].

To investigate the structure governing the ion transport mechanism for plant vacuolar-type NHX, the putative three-dimensional structure of rice vacuolar-type NHX1 (OsNHX1) was established through homology modeling and AlphaFold 3.0. As the first vacuolar NHX identified in rice, OsNHX1 plays a crucial role in the vacuolar membrane by sequestering and transporting high concentrations of Na^+^ and K^+^ from the cytoplasm into the vacuole. The abundance of this antiporter is a key determinant of salt tolerance in rice [35]. The OsNHX1 model structure was supported by hydrophobic characteristics and empirical and phylogenetic data. The topology of the predicted OsNHX1 model contains thirteen transmembrane segments, with N- and C-termini located on each side of the membrane, which is consistent with the latest studies involving other protein crystal structures of the CPA1 superfamily. NHX proteins typically contain critical charged and structurally related amino acid residues, which are essential for the binding and transport of cations and hydrogen ions [23]. Using the ConSurf web server and protein structure analysis, we identified all highly conserved charged residues, as well as several conserved structure-related residues. The functions of these amino acids were validated through site-directed mutagenesis and the yeast heterologous expression system in the salt-sensitive yeast strain AXT3. The experimental results and the protein model of OsNHX1 suggest that the pore domain of the vacuolar-type OsNHX1 features a transmembrane charge-compensated pattern similar to that of NHE1 while also displaying a distinct charge distribution on either side of the pore domain. Furthermore, the ion transport mechanism of vacuolar-type OsNHX1 shares similarities with those of NHE1 and OsSOS1, but it also demonstrates distinct differences. Interestingly, this study identified three mutations (E81Q, E313Q, and D323N) that enhance the sodium–potassium ion transport capacity of OsNHX1. In the rice NHX gene family, the vacuolar-localized OsNHX1 serves as the primary transporter for Na^+^ compartmentalization and plays a pivotal role in salt stress tolerance. Under normal physiological conditions, OsNHX1 also participates in K^+^ transport and accumulation, significantly contributing to grain growth regulation [19,36]. These mutations represent promising target sites for gene editing, offering potential avenues for the development of salt-tolerant rice varieties. Overall, our findings enhance our understanding of the structure–function relationship of key residues in plant vacuolar-type NHX proteins. Furthermore, they lay the groundwork for future investigations into the ion transport mechanisms of these proteins.

## 2. Materials and Methods

### 2.1. Yeast Strain, Medium, and Plasmids

*Saccharomyces cerevisiae* strains AXT3 (*Δena1-4::HIS3*, *Δnha1::LEU2*, and *Δnhx1::TRP1*) have been described previously [14]. Yeast cells were grown in YPD (1% yeast extract, 2% peptone, and 2% glucose) or SD media (0.67% yeast nitrogen base, 0.192% amino acid supplements without uracil, and 2% glucose); all reagents were procured from Sangon Biotech Co., Ltd. (Shanghai, China). The yeast expression plasmids pYPGE15 (PGK promoter, URA3) have been described elsewhere [20]. The DNA fragment corresponding to the OsNHX1 was amplified by PCR and inserted between the BamHI and KpnI sites in pYPGE15 (pYPGE15-OsNHX1). The DNA fragment corresponding to the OsNHX1 without the stop codon was fused at the N-terminus of enhanced green fluorescent protein (EGFP), and the resulting fusions were subcloned into pYPGE15 using the BamHI and EcoRI sites (pYPGE15-OsNHX1-EGFP).

### 2.2. Protein Structure Prediction

The protein structure of OsNHX1 was predicted by homology modeling and AlphaFold-predicted structures. The Pfam database identified the crystal structure of PaNhaP (pdb_00004cza) [27] and OsSOS1 (pdb_00008iwo) [31] as the best template for the homology modeling of OsNHX1. The pair alignment between PaNhaP, OsSOS1, and OsNHX1 was instrumental in the construction of a three-dimensional model of OsNHX1 using the SwissPdb viewer’s homology modeling program (https://swissmodel.expasy.org/, accessed on 23 October 2025). The AlphaFold-predicted structures of OsNHX1 were predicted by the AlphaFold protein structure database (https://alphafold.ebi.ac.uk/, accessed on 23 October 2025) [33,34]. These three-dimensional models of OsNHX1 were compared with the other Na^+^, K^+^/H^+^ antiporter protein crystal structures, such as PaNhaP (pdb_00004cza), NHE1 (pdb_00007dsv), and OsSOS1 (pdb_00008iwo) from the CPA1 family [27,29,31] and EcNhaA (pdb_00007s24), TtNapA (pdb_00004bwz), and NHA2 (pdb_00007b4l) from the CPA2 family [24,25,28]. The model structures were visualized using UCSF ChimeraX 1.9 software [37].

### 2.3. Evolutionary Conservation Analysis and Ion–Protein Interaction Analysis

The ConSurf web server was used to calculate evolutionary conservation scores via the Bayesian method (https://consurf.tau.ac.il, accessed on 23 October 2025) [38]. The AlphaFold 3.0 server was used to predict the way ions (Na^+^ and K^+^) and proteins interact based on OsNHX1 sequence data (https://alphafoldserver.com, accessed on 23 October 2025) [33,34].

### 2.4. Site-Directed Mutagenesis and Transformation of Yeast

Each OsNHX1 point mutation was created using PCR with the appropriate primer sets (Appendix A) and confirmed by sequencing. The full-length OsNHX1 and different mutations of OsNHX1 ORFs without the stop codon were separately fused at the N-terminus of EGFP, and the resulting fusions were inserted between the BamHI and KpnI sites in pYPGE15. These constructs were introduced into AXT3 cells via a typical PEG/LiAc protocol as described previously [17]. For the yeast functional assay, the point mutants of OsNHX1 were introduced into the yeast mutant strain AXT3. Empty pYPGE15-EGFP was used as a negative control. pYPGE15-OsNHX1 and pYPGE15-OsNHX1-EGFP were used as positive controls.

### 2.5. Preparation of Yeast Microsomal Membranes and Western Blotting

AXT3 cells that were endowed with the appropriate pYPGE15 plasmids were cultivated in 5 mL of SD medium at 30 °C until reaching saturation. Thereafter, 5 mL of this preculture was transferred into 500 mL of SD medium at 30 °C until reaching saturation. The culture was subsequently inoculated into 3000 milliliters of YPD medium and cultivated overnight. The yeast cells were harvested, washed, and broken in 20% glycerol (Sangon), 1 mM dithiothreitol (Sangon), 2 mM EDTA (Sangon), 100 mM Tris-HCl (pH 7.5) (Sangon), 1 mM phenylmethylsufonyl fluoride (Sangon), and 1 mM protease inhibitor cocktail (Sigma) by vortexing with glass beads. The lysate was centrifuged at 30,000× *g* for 30 min, and the supernatant was centrifuged at 100,000× *g* for 90 min to obtain the microsomal fraction. Centrifugation was performed using a Beckman Coulter Optima^TM^ XPN-80 ultracentrifuge (Brea, CA, USA). The proteins (25 µg) were separated by SDS-PAGE and then subjected to Western blotting using the anti-GFP antibody from Beyotime Biotech Inc. (Beyotime) (Shanghai, China), and pYPGE15-OsNHX1-EGFP was used as a positive control. ImageJ (64-bit Java 8) (USA) was used for the readings from the Western blotting.

### 2.6. Mutant Functional Assays

The functional assays of OsNHX1 point mutation were performed by the drop tests. In the drop tests, yeast cells were first cultivated in YPD (1% yeast extract, 2% peptone, and 2% glucose) and then transferred to selective AP medium (8 mM phosphoric acid, 0.2 mM CaCl_2_, 10 mM L-arginine, 2 mM MgSO_4_, 2% glucose plus vitamins and trace elements, adjusted to pH 5.8 with phosphoric acid) at 30 °C until they had become saturated; all reagents were procured from Sangon Biotech Co., Ltd. The saturated cultures were diluted to an OD600 of 0.5 ± 0.01, and then 4 μL volumes of several serial (10^−1^) dilutions were spotted on AP plates supplemented with NaCl or KCl at pH 5.8 and YPD plates containing hygromycin B (HYG) (Sangon).

## 3. Results and Discussion

### 3.1. The 3D Model of OsNHX1 Shares a Typical “Funnel” Fold

The protein structure of OsNHX1 was predicted based on the crystal structure of PaNhaP (pdb_00004cza) and OsSOS1 (pdb_00008iwo) as the best template for the homology modeling of OsNHX1, and the AlphaFold-predicted structure of OsNHX1 was predicted by the AlphaFold protein structure database (Figure 1A). Interestingly, the OsNHX1 protein structure predicted using homology modeling exhibited a high degree of similarity to the OsNHX1 protein structure predicted by AlphaFold 3.0 (Figure 1B). The range of transmembrane (TM) 1 of AlphaFold-predicted structures was longer than that of the other homology modeling structures. Nevertheless, the regions of TM1 predicted by AlphaFold 3.0 and those of AtNHX1 identified by experiments were essentially the same (Figure 1B) [20], as suggested by the precision of AlphaFold’s predicted structure. The three-dimensional (3D) models of OsNHX1 revealed a characteristic “funnel” structure, which is associated with the alternating-access mechanism core for Na^+^/H^+^ exchange observed in PaNhaP, NHE1, and OsSOS1 [27,29,31]; this structure is formed by transmembrane helices TM5–12. Nevertheless, the typical “funnel” structures are assembled by TM4–11 in NHA2 [28], EcNhaA [23,24], and TtNapA [25]. In the OsNHX1 model, the TM5 and TM12 segments unfold to form extended peptides at the center of the helix, where they intersect with each other. Additional helices are positioned centrally within the membrane. The TM5–12 assembly is situated between TM3, TM5, TM6, TM10, and TM12 helices. The external funnels are shaped by TM2, TM4, TM8, TM9, and TM13 helices (Figure 1A). In summary, the OsNHX1 model is consistent with the typical “funnel” fold as a structural basis for the alternating-access mechanism.

### 3.2. The Predicted Topology of OsNHX1 Contains 13 Transmembrane Segments

The topological models of plant vacuolar-type NHX have been previously described for AtNHX1 [20,32]. Yamaguchi et al. proposed a topological model of AtNHX1 that contains 12 transmembrane segments, with N- and C-termini located at the cytoplasmic side and vacuolar lumen side, respectively. According to the proposed model, the hydrophobic domain of AtNHX1 consists of nine transmembrane segments and three membrane-associated hydrophobic regions (TM3, TM5–6) that do not cross the membrane entirely. Sato and Sakaguchi proposed a topological model of AtNHX1 that contains 12 transmembrane segments, with both N- and C-termini located at the cytoplasmic side. According to the proposed model, the hydrophobic domain of AtNHX1 consists of twelve transmembrane segments and an intramembrane loop (H10) that do not cross the membrane entirely.

In this study, using homology modeling and AlphaFold-predicted structures, the topological model of plant vacuolar-type NHX (OsNHX1) was generated (Figure 1B,C), and the topology of the predicted OsNHX1 model was compared with AtNHX1, which was described by Yamaguchi et al. (Figure 1B). According to the proposed model, OsNHX1 contains thirteen transmembrane segments, with N- and C-termini located at the cytoplasmic side and vacuolar lumen side, respectively. The hydrophobic domain of OsNHX1 consists of eleven transmembrane segments and two transmembrane segments (TM8–9) that do not cross the membrane entirely. The range of TM8–9 is generally consistent across all three predicted models of OsNHX1, which were the key transmembrane segments constituting the typical “funnel” fold (Figure 1A). The typical “funnel” fold formed by the two non-expanded helices of TM8 and 9 is also highly conserved in the crystal structures of NHE1 [29], PaNhaP [27], and OsSOS1 [31]. However, Yamaguchi et al. proposed a topological model of AtNHX1 in which the range of TM8–9 was merged to form TM10 [20]. Yamaguchi et al.’s study may be questioned based on the vacuole protease protection assays employed, which demonstrated an absence of discriminatory capacity with respect to regions of incomplete membrane expansion [20]. Sato and Sakaguchi’s study may be questioned due to its use of protein fragments rather than the full-length protein [32]. The topology of the predicted OsNHX1 model contains thirteen transmembrane segments, with N- and C-termini located on each side of the membrane, which is more consistent with the latest studies with other protein crystal structures of the CPA1 superfamily, such as NHE1 [29], PaNhaP [27], and OsSOS1 [31].

### 3.3. The OsNHX1 Model Structure Is Consistent with Hydrophobic Characteristics, the “Positive-Inside” Rule, and Evolutionary Conservation

The structures of membrane proteins exhibit distinct common features in the distribution of positively charged and hydrophobic residues. In this study, the compatibility of the OsNHX1 model with these generic characteristics was evaluated (Figure 2 and Figure 3A). The OsNHX1 model exhibited the characteristic physicochemical properties of transporter proteins. The most polar residues were found to be concentrated in the core or on extramembrane loops, while the hydrophobic residues were oriented towards the membrane (Figure 2).

Previous research has identified a distinctive pattern in the distribution of positively charged amino acids within membrane proteins. Specifically, the intracellular regions of these proteins consistently exhibit a higher abundance of positively charged residues compared with their extracellular counterparts. This empirical observation, commonly referred to as the “positive-inside” rule, provides a useful framework for assessing the topological organization of membrane proteins [39,40]. This distribution has been readily demonstrated in the protein crystal structure of NHE1 [29], PaNhaP [27], OsSOS1 [31], NHA2 [28], EcNhaA [23,24], and TtNapA [25]. An analysis of the OsNHX1 3D model, using the PaNhaP model as the template, revealed nine arginine/lysine residues on the cytoplasmic side of the membrane and sixteen on the luminal side (Figure 3A). An analysis of the OsNHX1 3D model, using the OsSOS1 model as the template or AlphaFold 3.0, revealed nine arginine/lysine residues on the cytoplasmic side of the membrane and seventeen on the luminal side (Figure 3A). These results suggest that the OsNHX1 model structure was consistent with hydrophobic characteristics and the “positive-inside” rule.

Using the ConSurf web server, the evolutionary conservation score of the OsNHX1 model was calculated (Figure 3B). The value is approximately larger as the conservatism increases. The results indicate that the core region of the OsNHX1 model was highly conserved, whereas the loop region outside the transmembrane region was less conserved (Figure 3B). The core region of the OsNHX1 model revealed a characteristic “funnel” structure, which is formed by transmembrane helices TM5-12. Interestingly, the results suggest that some amino acid sites in the unconserved loop region outside the transmembrane region were highly conserved and may be responsible for forming the pocket region of the intermediate pore of ion channel proteins.

### 3.4. Experimental Validation of the OsNHX1 Model by Structure-Guided Design of Mutations

Previous studies have shown that NHX proteins have key charged amino acid sites, and structurally related amino acid sites play a crucial role in binding and translocating cations and hydrogen ions [23]. Using the ConSurf web server and protein structure analysis, we identified all highly conserved charged residues, along with several conserved structure-related residues, within the OsNHX1 model, including the conserved negatively charged residues Glu43, Glu50, Glu81, Asp147, Asp159, Asp170, Glu171, Glu182, Asp187, Asp250, Glu252, Glu268, Glu313, Asp323, and Asp326; the conserved positively charged residues Arg46, Lys101, Lys102, Lys103, Lys240, Arg246, Arg251, His287, Arg355, Arg392, and Lys438; and the conserved structure-related residues Pro90, Pro91, Thr158, Ser160, and Ser273. These residue distributions encompassed either the conservative core or the non-conservative extramembrane loops (Appendix A). To obtain structural and functional insights on OsNHX1, a site-directed mutagenesis of these highly conserved residues was undertaken. The glutamic and aspartic acids were substituted with glutamine and asparagine, respectively. The positively charged residues were substituted with alanine. This study was undertaken to investigate the correlations between the function and protein stability of these mutants. To this end, mutagenesis was performed using the PCR method, with OsNHX1-EGFP (EGFP fused to the C-terminus of OsNHX1; Appendix A shows that the EGFP fusion proteins did not modify the OsNHX1 function) as the template, and EGFP served as a negative control and did not differ from AXT3 yeast cells. To assess the functional impact of the mutants, AXT3 yeast cells transformed with each mutant were cultured in medium supplemented with varying concentrations of NaCl, KCl, or hygromycin B (Figure 4, Figure 5 and Figure 6). Western blotting showed that each functional mutant of OsNHX1-EGFP exhibited a similar expression level to the wild-type OsNHX1-EGFP (Figure 4D, Figure 5D and Figure 6D).

Figure 4 shows that the negatively charged amino acid mutants exhibited a variety of functional alterations. Under hygromycin B stress (Figure 4A), except for the mutants E268Q and E313Q, the other mutants exhibited altered functionality compared with wild-type OsNHX1. The mutants E43Q, E50Q, D147N, D159N, D170N, E171Q, E182Q, D187N, D250N and D326N did not show any tolerance to hygromycin B, while the mutant E252Q showed subtle decreases in tolerance under hygromycin B. The mutants E81Q and D323N showed significant increases in tolerance under hygromycin B. The susceptibility of AXT3 to hygromycin B toxicity is attributable to defective trafficking to vesicles [41]. The NHX proteins can partially compensate for hygromycin B sensitivity in AXT3 yeasts [42]. The results indicated that, except for Glu268 and Glu313, the other conserved negatively charged amino acids are essential for the function of OsNHX1 in the trafficking to the vesicles. Under NaCl stress (Figure 4B), the mutants E268Q and E313Q demonstrated no alteration in function in comparison with wild-type OsNHX1. The mutants E50Q, D147N, D159N, D170N, E182Q, D187N, D250N, and D326N did not show any tolerance to NaCl, while the mutants E43Q, E171Q, and E252Q showed subtle decreases in tolerance under NaCl. The mutants E81Q and D323N showed significant increases in tolerance under NaCl. Under KCl stress (Figure 4C), the mutant E268Q demonstrated no alteration in function in comparison with wild-type OsNHX1. The mutants D147N, D159N, E182Q, D187N, and D326N did not show any tolerance to KCl, while the mutants E43Q, E50Q, D170N, E171Q, D250N, and E252Q showed subtle decreases in tolerance under KCl. The mutants E81Q, E313Q and D323N exhibited enhanced tolerance to KCl, with the mutant E313Q demonstrating a particularly significant increase in tolerance. The results indicated that, except for Glu268, the other conserved negatively charged amino acids are essential for the function of OsNHX1 in Na^+^/K^+^ ion binding and translocation. Interestingly, the loss of the negative charge in Glu81 and Asp323 can enhance the Na^+^/K^+^ ion transport capacity of OsNHX1; however, the loss of the negative charge in Glu313 can only enhance the K^+^ ion transport capacity of OsNHX1. Glu81, Glu313, and Asp323 can serve as target sites for modifying OsNHX1 to enhance its functionality.

Figure 5 shows that the positively charged amino acid mutants exhibited a variety of functional alterations. Under hygromycin B and KCl stress (Figure 5A,C), the mutants R46A, K101A, K102A, K103A, K240A, and H287A did not show any tolerance to hygromycin B and KCl. The mutants R246A, R251A, R355A, and R392A demonstrated a significant reduction in functional activity relative to the wild-type OsNHX1. In contrast, the mutant K438A exhibited only minor decreases in tolerance when subjected to hygromycin B and KCl stress. Under NaCl stress (Figure 5B), the mutants H287A and K438A did not show any tolerance to NaCl. Similarly to the above, the mutants R246A, R251A, R355A, and R392A exhibited a marked decrease in functional activity compared with the wild-type OsNHX1. In contrast, the mutant R46A, K101A, K102A, K103A, and K240A exhibited only minor decreases in tolerance when subjected to NaCl stress. The results indicated that Arg246, Arg251, Arg355 and Arg392 are more essential for OsNHX1 protein function than the other conserved positively charged amino acids.

Figure 6 shows that the functions of the conserved structure-related residues Pro90, Pro91, Thr158, Ser160, and Ser273 in OsNHX1 were investigated. Two highly conserved prolines, i.e., Pro90 and Pro91 of TM3, were detected in OsNHX1. Other CAP families have also been described as having conserved proline at equivalent positions. Pro167 and Pro168 of TM3 in NHE1, as well as Pro82 of TM3 in OsSOS1, are involved in the conformational changes in the typical “funnel” fold in the cation transporter [29,31]. Pro90 and Pro91 of OsNHX1 were substituted with alanine. The mutants P90A and P91A showed significantly reduced activity under NaCl, KCl, and hygromycin B stress, compared with wild-type OsNHX1 (Figure 6). The results indicate that the functions of Pro90 and Pro91 in OsNHX1 were conserved, and the plant vacuolar-type NHX (OsNHX1) shared a similar alternating-access mechanism with NHE1, but it differs from the plant plasma membrane-type NHX (OsSOS1). The highly conserved Thr158 of TM5 in OsNHX1 was investigated in OsNHX1. The equivalent positions of the other CAP family have also been described—Thr132 of TM4 in EcNhaA [24], Thr126 of TM4 in TtNapA [25], Thr131 of TM5 in MjNhaP1 [26], and Thr129 of TM5 in PaNhaP [27]—which are involved in ion coordination. Thr158 of OsNHX1 was substituted with alanine. The mutant T158A exhibited significantly reduced activity under stress conditions induced by NaCl, KCl, and hygromycin B compared with the wild-type OsNHX1 (Figure 6). The results indicated that the functions of Thr158 in OsNHX1 were conserved. Ser160 (TM5) in OsNHX1 was exclusively conserved in the plant vacuolar-type NHX. There are no equivalent positions in the other CAP family, but Ser160 is adjacent to Thr158 and Asp159, which are functionally conserved amino acids in OsNHX1 (Figure 4 and Figure 6). The highly conserved Ser273 (TM8/9) in OsNHX1 is equivalent in position to Ser351 (TM8/9) in NHE1 and Ser258 (TM8/9) in OsSOS1, respectively, and serves as the main cation binding site [29,31]. To investigate the impact of spatial site resistance and charge on these two conserved serines, the Ser160 and Ser273 of OsNHX1 were independently substituted with alanine, proline (strong helix breaker), aspartic acid (negatively charged residue), and lysine (positively charged residue). The mutants S160A, S160P, S160D, and S160K showed significantly reduced activities under NaCl, KCl and hygromycin B stress compared with wild-type OsNHX1 (Figure 6). Interestingly, the mutants S160K and S160P were more sensitive to salt and hygromycin B stress than the mutants S160A and S160D, and the mutants S160D were more sensitive to NaCl stress than the mutants S160A (Figure 6). Previous studies have shown that alanine has the smallest residual group, but unlike proline, it does not disrupt the helix [43]. Ser160 of OsNHX1 is located at the bend of the TM5 helix. The results indicated that both the charge state and spatial site resistance can affect the functionality of Ser160, yet the disruption of the TM5 helix and positive charges are more likely to affect its functionality. The mutants S273D and S273K showed significantly reduced activities under NaCl and hygromycin B stress, and the mutant S273K showed significantly reduced activities under KCl stress compared with wild-type OsNHX1 and the other Ser273 mutants (Figure 6). Ser273 of OsNHX1 is located between the TM8 and 9 helices. The results indicated that spatial site resistance and helix breaking cannot affect the functionality of Ser273. The charge state of Ser273 is a pivotal determinant of its functionality.

### 3.5. OsNHX1 Shares a Similar Transmembrane Charge-Compensated Pattern with Nhe1

In this study, we employed site-directed mutagenesis to investigate the ion transport activity of all highly conserved charged residues. Several functionally critical amino acids were identified (Figure 4, Figure 5 and Figure 6). Functional experimental results and the OsNHX1 model suggested a charge-compensated pattern similar to CPA1 family members (Figure 7A), such as NHE1 [29], PaNhaP [27], and OsSOS1 [31] (Figure 7B), but different from CPA2 family members, such as EcNhaA [24], NHA2 [28], and TtNapA [25] (Figure 7C). Three conserved, negatively charged amino acids, i.e., Asp159 (TM5), Glu182 (TM6), and Asp187 (TM6), were detected in OsNHX1. Other CPA1 family members, i.e., NHE1, PaNhaP, and OsSOS1, have also been reported to possess conserved basic residues at analogous positions, i.e., Asp238 (TM5), Glu262 (TM6), and Asp267 (TM6) in NHE1; Asp130 (TM5), Glu154 (TM6), and Asp159 (TM6) in PaNhaP; and Asp147 (TM5), Glu171 (TM6), and Asp176 (TM6) in OsSOS1. The Asp187 and the Asn186 in TM6 of OsNHX1 form the ND motif, which is highly conserved among members of the CPA1 family and serves as a critical structural domain for ion binding and transport, such as N266-D267 in NHE1 [29], N158-D159 in PaNhaP [27], and N175-D176 in OsSOS1 [31]. In the CPA2 family, the ND motif is replaced by a DD motif, such as D163-D164 in EcNhaA, D277-D278 in NHA2 [28], and D156-D157 in TtNapA [25]. Three conserved, positively charged amino acids were detected, i.e., Arg355 (TM11) and Arg392 (TM12) in OsNHX1, corresponding to Arg425 (TM11) and Arg458 (TM12) in NHE1 [29]; Arg337 (TM11) and Arg362 (TM12) in PaNhaP [27]; and Arg341 (TM11) and Arg373 (TM12) in OsSOS1 [31]. Interestingly, the Glu313 (TM10) in OsNHX1 is not highly conserved within the CPA1 family members, unlike Asp159, Glu182, Asp187, Arg355, and Arg392 in OsNHX1. The Glu313 in OsNHX1 corresponds to a site that is uniquely found in NHE1 (Glu391 in TM10). No analogous site was identified in either PaNhaP or OsSOS1 (Figure 7B). However, functional experimental results indicate that Glu313 in OsNHX1 serves as a potassium ion binding site (Figure 4), whereas no studies have demonstrated a similar role for Glu391 in NHE1. These results indicate that OsNHX1 shares a similar transmembrane charge-compensated pattern with NHE1, but there are differences between the two transporters regarding function and ion transport mechanisms.

This study also predicts the interactions between amino acids within the active site of OsNHX1 (Figure 8). The three 3D models of OsNHX1 revealed a similar pattern of amino acid interactions, which were the interaction of Arg355 with Glu182, the interaction of Glu182 with Asn186, and the interaction of the ND motif (Asn186 and Asp187) with Thr158 and Ser160 (Figure 8A). However, OsNHX1 demonstrates a similar, yet distinct, pattern of amino acid interactions when compared with other CPA1 family members, such as NHE1, PaNhaP, and OsSOS1 (Figure 8B). Like the interaction of Arg355 with Glu182 in OsNHX1, similar interactions were observed in NHE1 (the salt bridge with Arg425 and Glu262), PaNhaP (the interaction of Arg337 with Glu154), and OsSOS1 (the interaction of Arg341 with Glu171) [27,29,31]. In contrast to the interaction between Glu182 and Asn186 in OsNHX1, a conserved serine interacts with the ND motif in NHE1 (the interaction of Ser263 with Asn266 and Asp267), PaNhaP (the interaction of Ser155 with Asn158 and Asp159), and OsSOS1 (the interaction of Ser172 with Asn175 and Asp176). Previous studies have demonstrated that the polar sidechain of the conserved serine appears critical for ion transport in NHE1 (Ser263) [29], PaNhaP (Ser155) [27] and OsSOS1 (Ser172) [31], but this conserved serine site is not present in OsNHX1 (Figure 8). The conserved threonine and aspartate residues in TM5 form a TD motif in PaNhaP (Thr129 and Asp130) [27] and OsSOS1 (Thr146 and Asp147) [31], which is critical for Na^+^ binding. In PaNhaP (OsSOS1), the conserved threonine Thr129 (Thr146) can interact with Ser155 (Ser172), Asn158 (Asn175) and Asp159 (Asp176), respectively (Figure 8B). However, Thr158, Asp159, and Ser160 form a TDS motif in OsNHX1, which interacts with the ND motif (Asn186 and Asp187) (Figure 8A). Functional experimental results indicate that Ser160 in OsNHX1 is critical for ion transport (Figure 6), but no analogous serine residues were detected in NHE1, PaNhaP, and OsSOS1. These results indicate that the vacuolar-type OsNHX1 exhibits a distinct ion transport mechanism that differs from that of the other CPA1 family members.

### 3.6. OsNHX1 Contains a Unique Charge Distribution on Both Sides of the Pore Domain

In this study, several functionally critical charged residues have been identified (Figure 4, Figure 5 and Figure 6). While some amino acids are situated within the transmembrane (TM) regions (Figure 7A), most functional sites are located on either side of the pore transmembrane domains (Figure 9). Figure 9 shows that the functionally critical charged residues Glu81, Asp147, Asp323, and Asp326 were located within the non-conservative extramembrane loops on the cytoplasmic side, while the Glu43, Glu50, Asp170, Glu171, Arg246, Asp250, Arg251, and Glu252 residues were located within the non-conserved extramembrane loops on the vacuolar lumen side. The distribution of these amino acids exhibits distinct patterns, with the majority of negatively charged amino acids localized near the pore domain (indicated by the yellow circle in Figure 9). Notably, Glu81 and Asp147 were positioned on the cytoplasmic side of the pore domain, whereas Glu43, Glu50, Asp170, and Glu171 were situated on the vacuolar lumen side. The evidence indicates that the negatively charged residues adjacent to the loop at both termini of the core structural domain play a significant role in modulating the ion transport function of OsNHX1. Additionally, on the cytoplasmic side, four functionally critical charged residues, including Arg246, Asp250, Arg251, and Glu252, form a functional domain (indicated by the green circle in Figure 9), while on the vacuolar lumen side, Asp323 and Asp326 were located within a region of concentrated negative charges (indicated by the purple circle in Figure 9). Based on our experimental findings (Figure 4, Figure 5 and Figure 6), mutations occurring on the cytoplasmic side (E81Q and D323N) of OsNHX1 enhance its ion transport capacity, whereas mutations on the vacuolar side invariably lead to a loss of function. Vacuolar-type NHX facilitates the transport of sodium and potassium ions from the cytoplasm into the vacuole, with these ions binding on the cytoplasmic side and protons binding on the vacuolar side. These results suggest that Glu81 may play a role in sodium and potassium ion binding. Furthermore, the functionally critical charged residues on the vacuolar side may affect proton binding and transport. Similar phenotypes were observed in the NHE1, where Arg180, Arg327, Glu330, and Arg440 are located in extramembrane loops implicated in pH regulation [44].

### 3.7. The Vacuolar-Type OsNHX1 Exhibits a Distinct Ion Transport Mechanism

In this study, the Na^+^ and K^+^ ion binding sites of OsNHX1 were predicted by AlphaFold 3.0 (Figure 10A). The results indicate that Thr158, Asn186, and Asp187 constitute the sodium ion binding site, while Ser51, Asp159, and Glu313 form the potassium ion binding site. Based on our experimental findings (Figure 4, Figure 5 and Figure 6), the mutants E50Q, T158A, D159N, and D187N showed significantly reduced activities under NaCl, KCl, and hygromycin B stress, and the mutant E313Q exhibited enhanced tolerance to KCl compared with wild-type OsNHX1. The experimental results further corroborate the predictive accuracy of AlphaFold 3.0.

Based on our experimental findings and the structural alignment of proteins, we have identified critical amino acid residues that significantly influence ion binding and transport in OsNHX1 (Figure 10B). Detailed information about the specific residue positions can be found in Appendix A. The crystal structure of NHE1 and OsSOS1 showed that the Glu131, Asp172, Asp238, Glu262, Ser263, Asp267, and Glu391 of NHE1, as well as the Thr146, Asp147, Glu171, Ser172, Asp176, and Ser258 of OsSOS1, appear to be critical for ion transport [29,31]. In our OsNHX1 model, the Glu50 (TM2) of OsNHX1 corresponds to Glu131 (TM2) in NHE1, whereas this specific residue is not conserved in OsSOS1. In the NHE1 model, Asp172 (TM3) is located on the cytoplasmic side of Glu391 (TM10), which is crucial for NHE1 activity [45]. Glu87 (TM3) of OsSOS1 was detected at an equivalent position [31], and whether it serves a similar function remains unclear. However, no similar amino acid sites were identified in OsNHX1. In the OsSOS1 model, Thr146 and Asp147 in TM5 form a conserved TD motif, which is critical for ion binding. However, in the NHE1 model, only Asp238 (TM5) corresponds to Asp147 (TM5) in OsSOS1, which is indirectly involved in cation binding by mediating the direct coordination of the water molecule [29,46]. In our OsNHX1 model, Thr158 and Asp159 in TM5 correspond to Thr146 and Asp147 of OsSOS1, but Ser160 is not conserved in OsSOS1 and NHE1. Our findings demonstrate that Thr158, Asp159, and Ser160 constitute a TDS motif in OsNHX1, which is essential for its activity. This structural feature distinguishes OsNHX1 from OsSOS1 and NHE1. The conserved glutamic acid, Glu182 (TM6) of OsNHX1, Glu262 (TM6) of NHE1, and Glu171 (TM6) of OsSOS1, were detected at equivalent positions (Figure 10B). The residues Glu262 (TM6) and Arg425 (TM11) in NHE1, as well as Glu171 (TM6) and Arg341 (TM11) in OsSOS1, form a salt bridge that influences protein function, and they do not directly participate in ion coordination [29,31]. A similar interaction of Glu182 with Arg355 was observed in OsNHX1 (Figure 8). This indicates that the Glu182 of OsNHX1 is functionally conserved. The conserved serines, Ser263 (TM6) of NHE1 and Ser172 of OsSOS1, were detected at equivalent positions (Figure 10B), functioning as binding sites for Na^+^ ions [29,31]. However, no similar amino acid sites were identified in OsNHX1. The conserved asparagine and aspartic acid residues in TM6 form an ND motif, which was detected in OsNHX1 (Gln186 and Glu187), NHE1 (Gln266 and Glu267), and OsSOS1 (Gln175 and Glu176), which is a critical motif for the alternating-access mechanism of cation translocation [29,31]. For a long time, the ND motif of the CPA1 family and the DD motif of the CPA2 family have been regarded as the crucial structural domains distinguishing the two families [23,47].

The conserved serine, Ser351 (TM8/9) of NHE1, serves as the cation binding site [44], and Ser273 (TM8/9) of OsNHX1 was detected at an equivalent position (Figure 10B). The experimental results demonstrate that the charge state of Ser273 is a crucial determinant of OsNHX1 functionality (Figure 6). However, in the vacuolar-type NHX protein MdNHX1 from apple (*Malus domestica* Borkh.), the corresponding residue is Ser275. Notably, phosphorylation of Ser275 has been shown to enhance the Na^+^/H^+^ transport activity of MdNHX1 [48]. As we know, aspartic acid is commonly employed to mimic the phosphorylation of serine in various biochemical studies [49]. The presence of a negative charge at this conserved position enhanced Na^+^/H^+^ transport activity in the vacuolar-type NHX protein MdNHX1 [48] but abolished the function of its counterpart OsNHX1 (Figure 6). These results suggest the presence of potential species-specific functional variations in this conserved serine residue. Moreover, Ser258 (TM8/9) of OsSOS1 was detected at an equivalent position [31], but whether it serves a similar function remains unclear. In the NHE1 model, Glu391 (TM10) is located at the membrane level, similar to D267 (TM10), which is functionally important [29]. Glu313 (TM10) of OsNHX1 was detected at an equivalent position (Figure 10B), which is crucial for the potassium ion binding site (Figure 4 and Figure 10A). The functional role of the conserved glutamic acid site in NHE1 exhibits significant divergence from that observed in OsNHX1. Moreover, no similar amino acid sites were identified in OsSOS1 (Figure 10B). Two conserved proline residues, Pro167 and Pro168 (TM3) of NHE1, play important structural roles in ion translocation [29]. In our OsNHX1 model, Pro90 and Pro91 (TM3) were detected at equivalent positions (Figure 10B), which is crucial for the OsNHX1 activity. Pro82 (TM3) of OsSOS1 was detected at an equivalent position [31], but whether it serves a similar function remains unclear. In summary, our comparative analysis of key amino acid sites responsible for ion transport in OsSOS1 and NHE1 reveals that the vacuolar-type OsNHX1 employs a unique ion transport mechanism. While this mechanism shares similarities with those of NHE1 and OsSOS1, it also exhibits distinct differences.

## 4. Conclusions

In this study, the three-dimensional structure of vacuolar-type NHX (OsNHX1) was established through homology modeling and AlphaFold 3.0. The OsNHX1 model structure is consistent with hydrophobic characteristics, the “positive-inside” rule, and evolutionary conservation. The topological models of plant vacuolar-type NHXs were described for the AtNHX1 by vacuole protease protection assays that contain 12 transmembrane segments, with N- and C-termini located at the cytoplasmic side and vacuolar lumen side, respectively. Unlike the previous AtNHX1 topology model, the topology of the predicted OsNHX1 model contains thirteen transmembrane segments, with N- and C-termini located on each side of the membrane. The primary distinction between the two topology models lies in the merging of the TM8–9 region to form TM10 in AtNHX1. According to our OsNHX1 model, the TM8–9 region does not form a complete transmembrane helix, rendering it undetectable in vacuole protease protection assays. This limitation may result in experimental inaccuracies. Our OsNHX1 model is consistent with the latest studies on other protein crystal structures of the CPA1 superfamily, such as NHE1, PaNhaP, and OsSOS1.

Moreover, this study validated the protein model of OsNHX1 through functional experiments, which identified a set of key charged amino acids critical to its function. Mapping these amino acids onto the OsNHX1 model revealed that its pore domain exhibits a transmembrane charge-compensated pattern similar to that of NHE1 while also displaying a distinct charge distribution on either side of the pore domain. Our comparative analysis of key amino acid sites responsible for ion transport in the crystal structure of OsSOS1 and NHE1 reveals that the vacuolar-type OsNHX1 employs a unique ion transport mechanism. Based on experimental results, protein structural alignments, and predictions of ion binding sites, we identified that Glu50, Thr158, Asp159, Ser160, Glu182, Asp187, Ser273, and Glu313 were involved in ion binding and transport. Thr158, Asn186, and Asp187 constitute the sodium ion binding site, whereas Glu50, Asp159, and Glu313 are responsible for the binding of potassium ions. Meanwhile, the ND motif is formed by Gln186 and Glu187, while Thr158, Asp159, and Ser160 constitute the TDS motif, both of which are critical for OsNHX1 activity. Interestingly, this study identified three mutations (E81Q, E313Q, and D323N) that enhance the sodium–potassium ion transport capacity of OsNHX1. These mutations present potential target sites for gene editing aimed at developing salt-tolerant rice varieties. Our OsNHX1 model advances the understanding of key structural and functional residues in plant vacuolar-type NHXs. Furthermore, it establishes a foundation for future investigations into the mechanisms underlying ion translocation, cation specificity and selectivity, and pH regulation in vacuolar-type NHXs.

## Figures and Tables

**Figure 1 biomolecules-15-01513-f001:**
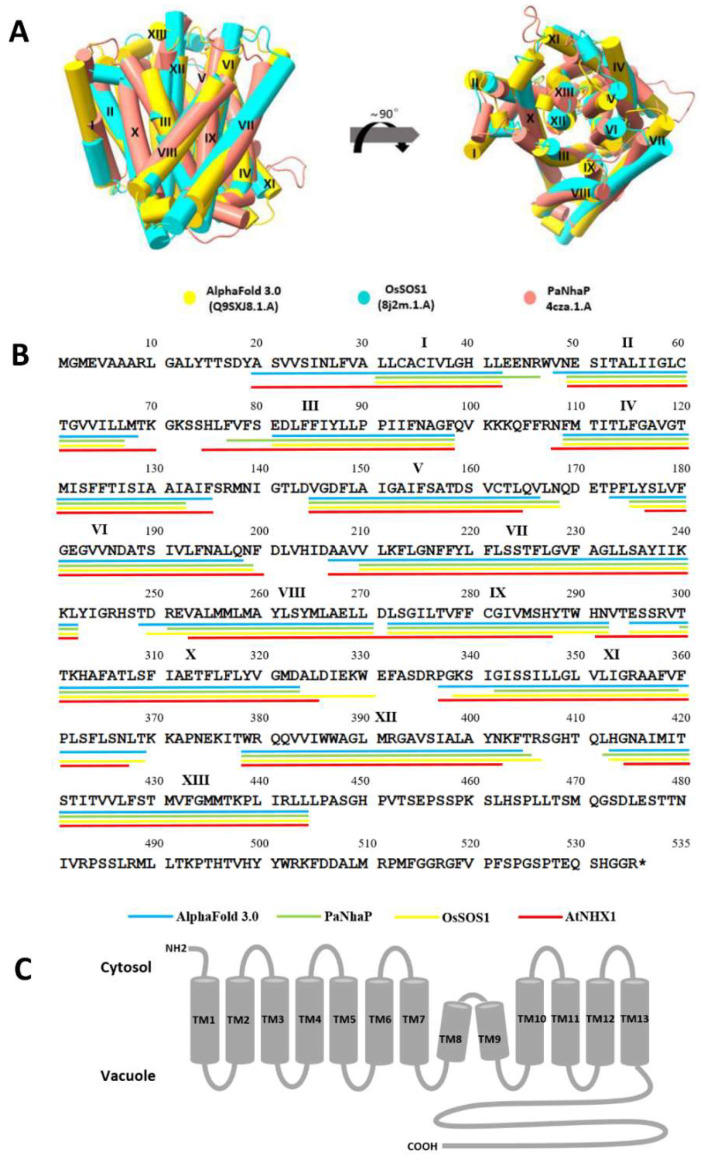
The membrane topology and three-dimensional (3D) model of OsNHX1. (**A**) The 3D model of OsNHX1 was predicted using homology modeling based on the crystal structure of PaNhaP (pink) and OsSOS1 (blue), supplemented by predictions from AlphaFold 3.0 (yellow). (**B**) The TM segments of the OsNHX1 sequence. The TM helices were predicted using homology modeling based on PaNhaP (green line) and OsSOS1 (yellow line), supplemented by predictions from AlphaFold 3.0 (blue line). The other TM helices were predicted based on the amino acid sequence alignment of AtNHX1 (red line). The asterisk denotes a stop codon. (**C**) The membrane topology model of OsNHX1 contains 13 transmembrane segments.

**Figure 2 biomolecules-15-01513-f002:**
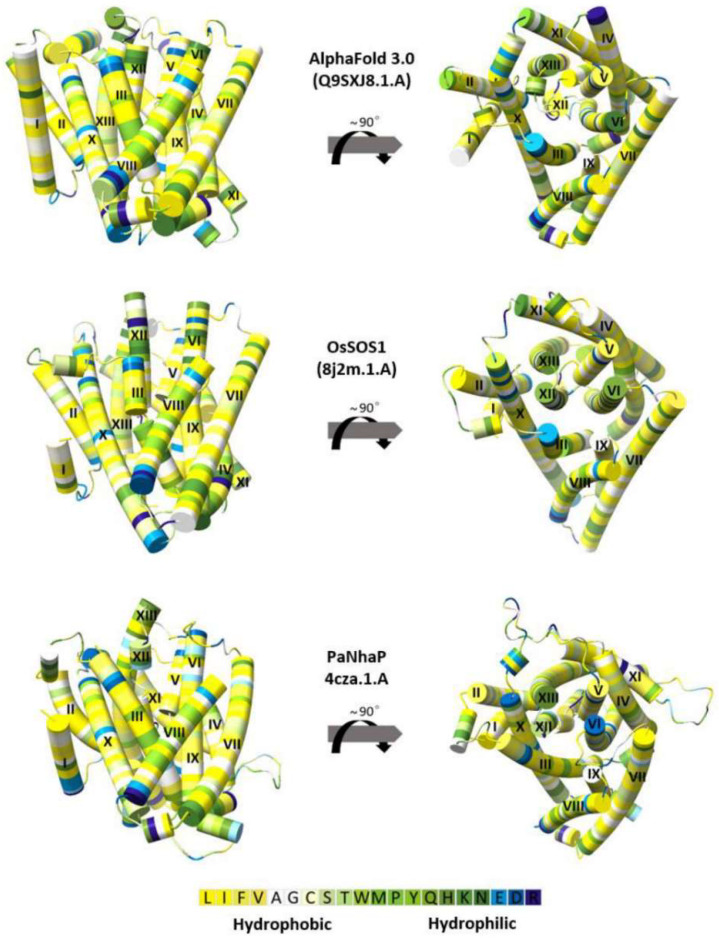
The hydrophobic properties of the OsNHX1 model. Residues are colored according to the hydrophobicity scale of Kessel and Ben-Tal [38]. Overall, the transmembrane (TM) helices of OsNHX1 are predominantly hydrophobic but contain polar and titratable residues.

**Figure 3 biomolecules-15-01513-f003:**
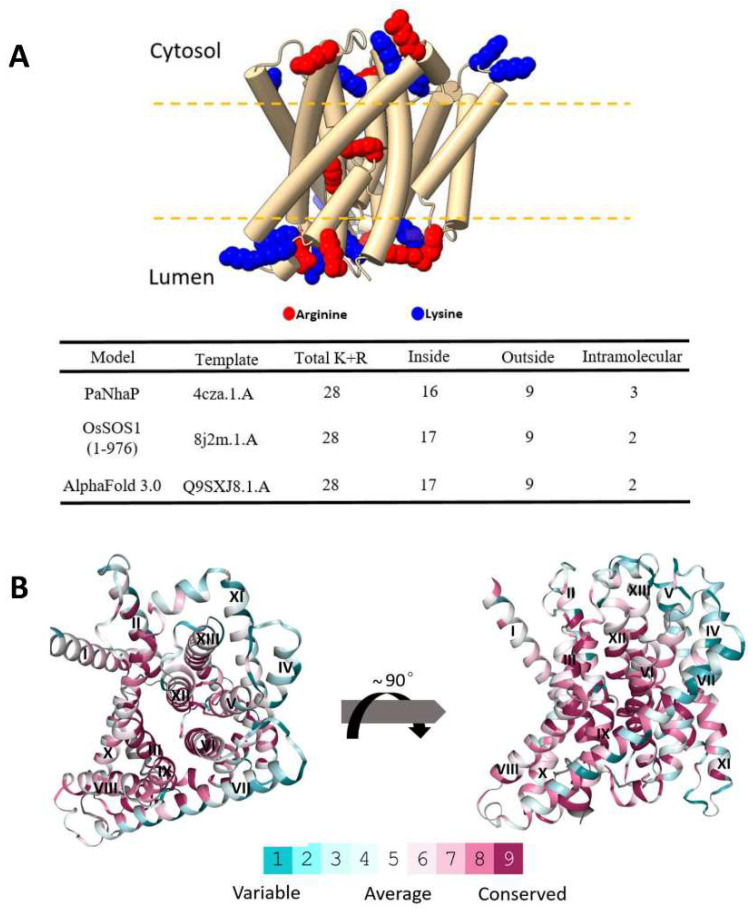
The OsNHX1 model structure is consistent with the “positive-inside” rule and evolutionary conservation. (**A**) The OsNHX1 model follows the “positive-inside” rule. The arginine and lysine residues in the OsNHX1 model are shown as red and blue spheres, respectively. The approximate boundaries of the hydrocarbon regions of the membrane are shown by yellow lines. (**B**) The structural model of OsNHX1 was validated based on evolutionary conservation. The structural model of OsNHX1 is depicted in both top and side views, with residues color-coded according to their ConSurf conservation scores [37].

**Figure 4 biomolecules-15-01513-f004:**
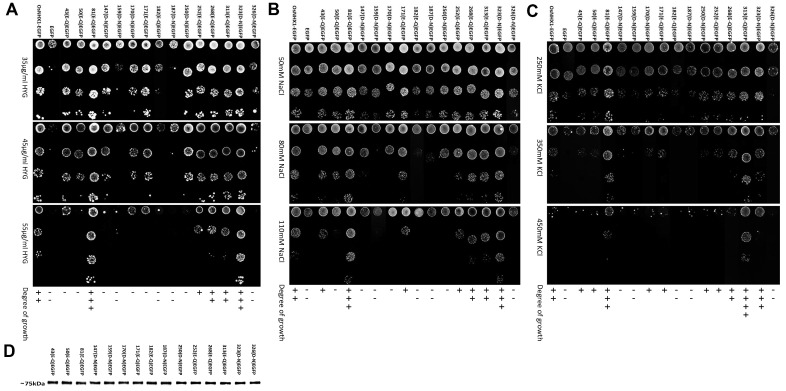
Phenotypic screening of the conserved negatively charged amino acid mutants of OsNHX1 in yeast. (**A**–**C**) The full-length OsNHX1 open reading frame (ORF), along with the negatively charged amino acid mutants of the OsNHX1 ORF lacking the stop codon, were independently fused to the N-terminus of EGFP and subsequently introduced into the salt-sensitive yeast strain AXT3. The saturated cultures were diluted to an OD600 of 0.5 ± 0.01, and then 4 μL volumes of 10-fold serial dilutions were applied onto YPD plates supplemented with hygromycin B (30, 45, and 55 μg/mL) (**A**) and AP plates supplemented with varying concentrations of NaCl (50, 80, and 110 mM) (**B**) and KCl (250, 350, and 450 mM) (**C**) at a pH of 5.8, respectively. (**D**) The protein expression level of the negatively charged amino acid mutants was detected by Western blotting. Empty pYPGE15-EGFP and pYPGE15-OsNHX1-EGFP were used as negative and positive controls, respectively. Original images can be found at Appendix A.

**Figure 5 biomolecules-15-01513-f005:**
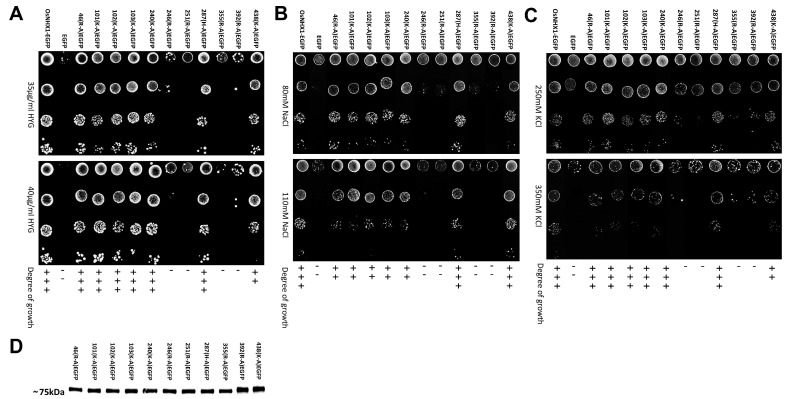
Phenotypic screening of the conserved positively charged amino acid mutants of OsNHX1 in yeast. (**A**–**C**) The full-length OsNHX1 open reading frame (ORF), along with the positively charged amino acid mutants of the OsNHX1 ORF lacking the stop codon, was independently fused to the N-terminus of EGFP and subsequently introduced into the salt-sensitive yeast strain AXT3. The saturated cultures were diluted to an OD600 of 0.5 ± 0.01, and then 4 μL volumes of 10-fold serial dilutions were applied onto YPD plates supplemented with hygromycin B (35 and 40 μg/mL) (**A**) and AP plates supplemented with varying concentrations of NaCl (80 and 110 mM) (**B**) and KCl (250 and 350 mM) (**C**) at a pH of 5.8, respectively. (**D**) The protein expression level of the positively charged amino acid mutants was detected by Western blotting. Empty pYPGE15-EGFP and pYPGE15-OsNHX1-EGFP were used as negative and positive controls, respectively. Original images can be found at Appendix A.

**Figure 6 biomolecules-15-01513-f006:**
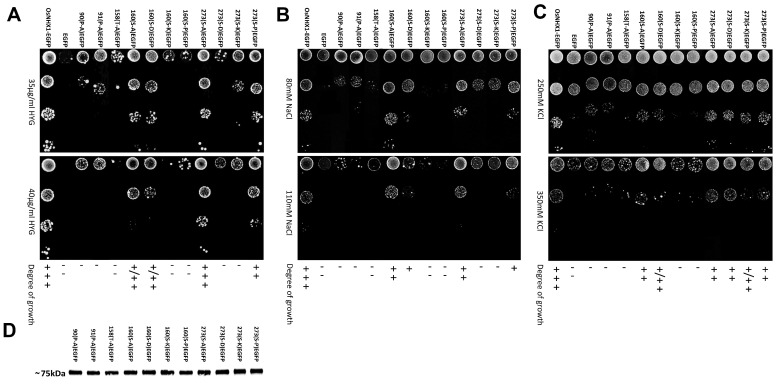
Phenotypic screening of the conserved structure-related amino acid mutants of OsNHX1 in yeast. (**A**–**C**) The full-length OsNHX1 open reading frame (ORF), along with the conserved structure-related amino acid mutants of the OsNHX1 ORF lacking the stop codon, was independently fused to the N-terminus of EGFP and subsequently introduced into the salt-sensitive yeast strain AXT3. The saturated cultures were diluted to an OD600 of 0.5 ± 0.01, and then 4 μL volumes of 10-fold serial dilutions were applied onto YPD plates supplemented with hygromycin B (35 and 40 μg/mL) (**A**) and AP plates supplemented with varying concentrations of NaCl (80 and 110 mM) (**B**) and KCl (250 and 350 mM) (**C**) at a pH of 5.8, respectively. (**D**) The protein expression level of the conserved structure-related amino acid mutants was detected by Western blotting. Empty pYPGE15-EGFP and pYPGE15-OsNHX1-EGFP were used as negative and positive controls, respectively. Original images can be found at Appendix A.

**Figure 7 biomolecules-15-01513-f007:**
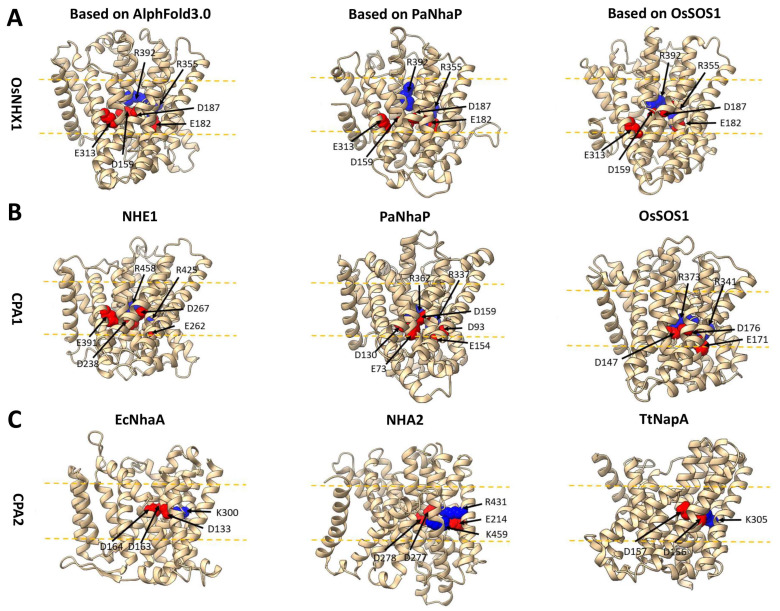
The transmembrane charge compensation in OsNHX1 and members of the CPA1 and CPA2 families. (**A**) The essential charged residues within the transmembrane regions of the three OsNHX1 models, which were predicted using AlphaFold 3.0 and homology modeling based on the crystal structure of PaNhaP and OsSOS1. (**B**,**C**) The essential charged residues within the transmembrane regions of the CPA1 (NHE1, PaNhaP, and OsSOS1) and CPA2 (EcNhaA, NHA2 and TtNapA) families. In all the panels, the negatively and positively charged residues are shown as red and blue spheres, respectively. The approximate boundaries of the hydrocarbon regions of the membrane are shown by yellow lines.

**Figure 8 biomolecules-15-01513-f008:**
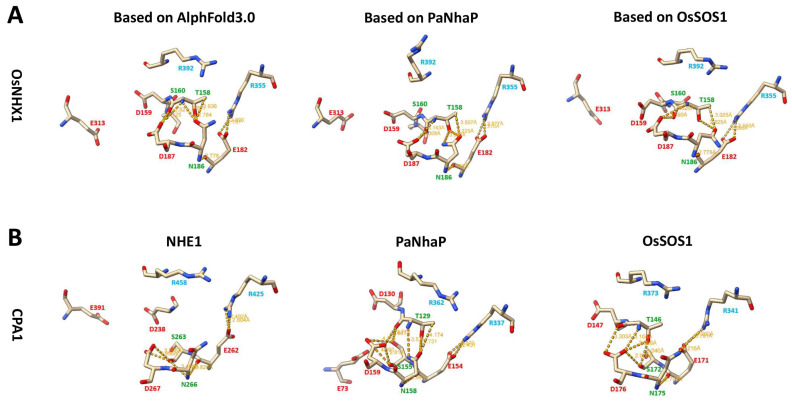
The predicted amino acid interactions in the active site of the OsNHX1 and CPA1 family members. (**A**) Representation of the side chains of amino acids that are hypothesized to influence the activity of OsNHX1, as inferred from the models derived from PaNhaP, OsSOS1 templates, and AlphaFold 3.0. (**B**) Representation of the active site side chains in CPA1 family members, including NHE1, PaNhaP, and OsSOS1. Yellow dashed lines denote potential interactions based on the proximity of residues.

**Figure 9 biomolecules-15-01513-f009:**
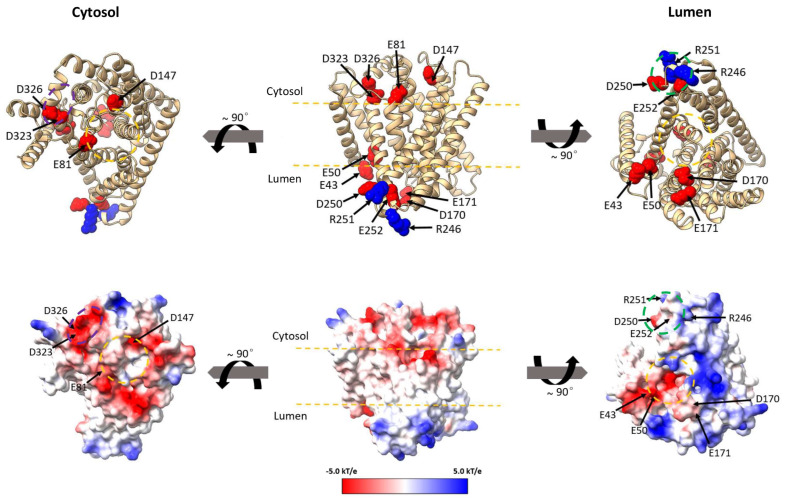
The distribution of functionally critical charged residues on both sides of the pore domain in OsNHX1. The protein structures are represented by ribbon diagrams (**top**) and surface area plots (**bottom**), respectively. On the left, a top view is presented from the cytoplasmic side of the membrane. In the middle, a side view is depicted parallel to the membrane, with the cytoplasmic side oriented upwards. On the right, a view is shown from the luminal side. The negatively and positively charged residues are shown as red and blue, respectively. The approximate boundaries of the hydrocarbon regions of the membrane are shown by yellow lines.

**Figure 10 biomolecules-15-01513-f010:**
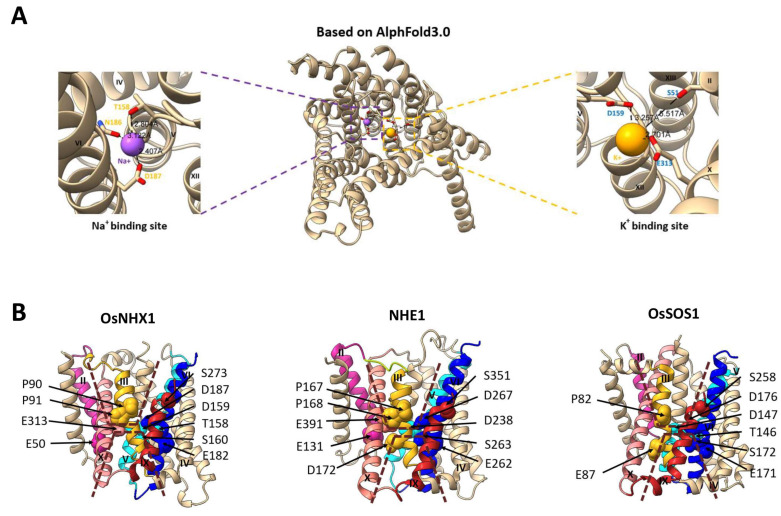
The vacuolar-type OsNHX1 exhibits a distinct ion transport mechanism. (**A**) The Na^+^ (left side) and K^+^ (right side) ion binding sites of OsNHX1 were predicted by AlphaFold 3.0. The distance between the amino acid and the ion is represented by the black dashed line. (**B**) The ion transport models of OsNHX1, NHE1, and OsSOS1. The cation-transport pathways of OsNHX1, NHE1, and OsSOS1 are composed of two discontinuous funnels formed by TM5–12, which is represented by the brown dashed line. The functional amino acids involved in ion binding and transport are identified in the OsNHX1, NHE1, and OsSOS1 models, respectively.

## Data Availability

Data are contained within the article and Appendix A.

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
