# Peer review of "Structural and Functional Characterization of the Vacuolar-Type Na+, K+/H+ Antiporter NHX1 from Rice (Oryza sativa L.)"

_biomolecules, 2025, doi:10.3390/biom15111513_

Round 1
Reviewer 1 Report
Comments and Suggestions for Authors
The submitted Manuscript provides interesting results about the modelled structure of rice NHX1 and seems publishable.
Revisions required or reject-resubmit.
1) The very abstract.
Abstract: Plant vacuolar-type Na+, K+/H+ antiporter (NHX) play important roles in pH, K+ homeostasis, and osmotic balance under normal physiological state.
Antiporter plays or antiporters play.
The language is not good in the whole text from the very abstract. Please, check and ask a native speaking colleague to correct for better readability.
2) The numbering of lines should be given.
3) Suggest to read and add more general and earlier references for sodium uptake mechanism by plants and groups of ion transporters, e.g.:
Pascal Mäser, Sébastien Thomine, Julian I. Schroeder, John M. Ward, Kendal Hirschi, Heven Sze, Ina N. Talke, Anna Amtmann, Frans J.M. Maathuis, Dale Sanders, Jeff F. Harper, Jason Tchieu, Michael Gribskov, Michael W. Persans, David E. Salt, Sun A Kim, Mary Lou Guerinot, Phylogenetic Relationships within Cation Transporter Families of Arabidopsis, Plant Physiology, Volume 126, Issue 4, August 2001, Pages 1646–1667, https://doi.org/10.1104/pp.126.4.1646
Apse MP, Aharon GS, Snedden WA, Blumwald E. Salt tolerance conferred by overexpression of a vacuolar Na+/H+ antiport in Arabidopsis. Science. 1999 Aug 20;285(5431):1256-8. doi: 10.1126/science.285.5431.1256. PMID: 10455050.
Apse MP, Blumwald E. Na+ transport in plants. FEBS Lett. 2007 May 25;581(12):2247-54. doi: 10.1016/j.febslet.2007.04.014. Epub 2007 Apr 18. PMID: 17459382
etc. going back to the very first publications and discoveries on the subject.
4) The quality of very good from the point of results and labour-intensive figures 4, 5 and 6 is extremely low.
5) The main part is devoted to modelling while the mechanism of ion transport is not always solved by crystal or cryo EM structures. The mechanisms of transport could be deciphered by combination of methods including electrophysiology, imaging at the level of molecules, structural methods with mutant amino acids etc.
6) All the figures with western blots should provide the protein ladders for molecular weights to see the sizes with indications of kDa. The original images should be provided.
7) Abstract and the main idea.
Furthermore, this study validated the OsNHX1 model via functional experiments, revealing a set of key charged amino acids essential for its activity.
Yeast complementation experiments were done; it’s the only functional assay completed. Heterologous expression in animal cells with electrophysiological characterization is required to reveal the activity of the mutated forms of the transporter.
8) Figure 3. The thickness of tonoplast membrane is about 7-10 nm than it 70-100 Angstrom. How could the transporter then fit the membrane? Are there any data how it is located and functions in rice tonoplast?
9) Modelling with lipids is required to see their roles in the functioning of the transporter and the effects on its structure. Specific modelling programs offer wide opportunities to do so, e.g. molecular dynamics software https://en.wikipedia.org/wiki/Molecular_dynamics
10) Figures 5, 6 and 4 are the most important ones, not the simple modelling results. They should be presented in the other way stressing their importance.
11) Potentially the stereo images could be given to present the molecules with their transport pathways.
12) Figure 9 – how the molecule fit the membrane when is 2.2 nm only?
13)What is the number of genes for NHX transporters in rice? Please, provide a few indications for rice plants in the introduction.
14) Methods.
Saccharomyces cerevisiae strains
In italics all the latin names.
+ methods should be described in more details. Origin of yeast strains, origin of antibodies, company for all the chemicals etc. + all the devices used.
15) The present Review suggests reject-resubmit to rewrite the paper for better presenting the good results. Potentially more experiments and modelling to be done.
Comments on the Quality of English LanguageThe very abstract.
Abstract: Plant vacuolar-type Na+, K+/H+ antiporter (NHX) play important roles in pH, K+ homeostasis, and osmotic balance under normal physiological state.
Antiporter plays or antiporters play.
The language is not good in the whole text from the very abstract. Please, check and ask a native speaking colleague to correct for better readability.
Reviewer 2 Report
Comments and Suggestions for Authors
Comments are included in the attached document.

Could be improved in the Materials and Methods section. It is OK in the other sections of the manuscript.
Round 2
Reviewer 1 Report
Comments and Suggestions for Authors
The Authors provided satisfactory responses to all the questions and comments of the present Reviewer and made the corresponding amendments that improved the Manuscript.
Still, some revisions are required.
1) The Review would present the nice results from figures 4-6 in a better way with larger scale of the figures. However, the question is for the Authors.
2) Figures 4-6. Please, indicate that EGFP served as a control and did not differ from AXT3.
3)
Lines 74-76.
[17], while its homolog AtNHX8 mainly transports Li+ [18]. In rice, the NHX gene family 74 have 6 members, including OsNHX1–4 (vacuolar membrane-type NHX), OsNHX5 (endo-75 membrane-type NHX) and OsSOS1 (plasma membrane-type NHX) [19]. 76
Gene family has 6 members, not has 6 members. Simple language problem.
4) Supplementary table S1. without the stop codon were fused the EGFP
Fused with or fused to.
5) Abstract: Plant vacuolar-type Na+, K+/H+ antiporters (NHXs) play important roles in pH, 16 K+ homeostasis, and osmotic balance under normal physiological conditions.
“pH and K+ homeostasis” is a better description.
6) The OsNHX1 model contains thirteen transmembrane segments and is sup-23 ported by hydrophobic characteristics, empirical and phylogenetic data. Furthermore, this 24
The correction is not correct. The earlier version was OK:
The OsNHX1 model contains thirteen transmembrane segments, which was supported by hydrophobic characteristics, empirical and phylogenetic data.
Alternatively:
The OsNHX1 model contains thirteen transmembrane segments according to hydrophobic characteristics, empirical and phylogenetic data.
Alternatively:
The OsNHX1 model contains thirteen transmembrane segments; the point is supported by hydrophobic characteristics, empirical and phylogenetic data.
7) Beckman Coulter OptimaTM XPN-80 197 ultracentrifuge
Country and state for the producer of the equipment.
8) Please, indicate which densitometer was used for the readings from the western blot.
9) The Review suggests major revision to fix the questions left and also suggests to find a native English colleague since the corrected language was corrected more to language over the sense at some revealed points. Then the Manuacript could be published, according to the opinion of the Reviewer.
Comments on the Quality of English LanguageSuggest to find a native English colleague since the corrected language was corrected more to language over the sense at some revealed points.
